# Temperature-Dependent Broadening of the Ultraviolet Photoelectron Spectrum of Au(110)

**DOI:** 10.3390/s21175969

**Published:** 2021-09-06

**Authors:** Tomonari Nishida, Ikuo Kinoshita, Juntaro Ishii

**Affiliations:** 1Graduate School of Nanobioscience, Yokohama City University, Yokohama 236-0027, Japan; n185219g@yokohama-cu.ac.jp; 2National Metrology Institute of Japan, National Institute of Advanced Industrial Science and Technology (AIST), Tsukuba 305-8563, Japan; j-ishii@aist.go.jp

**Keywords:** photoelectron spectroscopy, thermodynamic temperature, Fermi–Dirac distribution, broadening function, Fourier transform, energy resolution

## Abstract

To determine the thermodynamic temperature of a solid surface from the electron energy distribution measured by photoelectron spectroscopy, it is necessary to accurately evaluate the energy broadening of the photoelectron spectrum and investigate its temperature dependence. Broadening functions in the photoelectron spectrum of Au(110)’s surface near the Fermi level were estimated successfully using the relationship between the Fourier transform and the convolution integral. The Fourier transform could simultaneously reduce the noise of the spectrum when the broadening function was derived. The derived function was in the form of a Gaussian, whose width depended on the thermodynamic temperature of the sample and became broader at higher temperatures. The results contribute to improve accuracy of the determination of thermodynamic temperature from the photoelectron spectrum and provide useful information on the temperature dependence of electron scattering in photoelectron emission processes.

## 1. Introduction

Thermodynamic temperature is an essential parameter indicating the energy states of a system. In the field of surface science, thermodynamic analysis of atoms and molecules in a surface heat bath enables understanding of surface reactions in addition to static surface structures and electronic states. As with thermal designs of MEMS devices and developments of functional nanomaterials, it is necessary to focus on several surface layers to achieve thermal control for electronic devices [1], and heat generation states [2] and the microchannel heat sink in the cooling of microelectronic devices [3]. Therefore, it is important to measure the thermodynamic temperature and thermophysical properties localized on the surfaces.

Thermodynamic temperature is determined using the laws of physics derived from the statistical distribution of collective particles. Ideal gas particles in classical mechanics and bosons and fermions in quantum mechanics are governed by the Maxwell–Boltzmann (MB) distribution, Boson–Einstein (BE) distribution, and Fermi–Dirac (FD) distribution, respectively. The ideal gas law is derived from the MB distribution, and Planck’s radiation equation is derived from the BE distribution. As representative examples, gas thermometry and radiometric thermometry have already been developed as thermodynamic thermometry using their respective physical laws [4,5].

The FD distribution represents the occupancy of electrons in a density of states (DOS) at thermodynamic temperature. Ultraviolet photoelectron spectroscopy (UPS) measures the electron energy distribution in the valence band. In USP, a sample is irradiated with monochromatic light under an ultra-high vacuum (UHV), and the energy of electrons emitted from the sample surface is detected by an electron energy analyzer [6]. It is possible in principle to determine the thermodynamic temperature of a sample surface directly by measuring the electron energy spectrum based on UPS, subsequently normalizing the step function shape of the Fermi edge in the photoelectron spectrum by DOS, and then fitting the FD distribution function. We call this measurement technique photoelectron thermometry [7].

The electrons measured by UPS are limited to a few atomic layers on the surface. Therefore, this temperature measurement enables analysis of thermal phenomena localized on thin films and nano-surfaces, which are difficult to measure with existing temperature sensors. Moreover, since the FD distribution depends only on the thermodynamic temperature and not on the individual material property values, this measurement technique can be an absolute temperature measurement (primary thermometer) that requires no scale calibration against any reference thermometer. Further, while infrared radiation thermometers have a lower limit to practical measurement of around −30 °C, photoelectron emission has no restriction on the temperature value in principle. This technique enables a wider range of non-contact temperature measurements, including cryogenic region.

There have been a few reports of temperature measurements using UPS [8,9]. However, their temperature resolutions were as large as tens of K, and practical temperature accuracy (less than 1 K) has not been achieved. In our previous work, the electron energy distributions on the Cu(110) surface in the temperature range from 10 to 300 K were measured [10]. As a result of fitting the spectrum of each temperature with the convolution integral function of the Fermi–Dirac distribution function and the Gaussian function with the spectral broadening width as a fitting parameter, it was found that fitting with the same broadening width to spectra at various temperatures was impractical. The energy resolution is a function of the limitations of the electron energy analyzer and the light source, which can be regarded as constant in a series of measurements. On the other hand, factors that contribute to spectral broadening, such as sample surface conditions, excited state lifetime, and electron-lattice scattering in the process of electron emission can be dependent on temperature. It is necessary to determine the thermodynamic temperature and broadening width at one time in an individual spectrum of UPS.

In our recent research, high-energy resolution photoelectron spectroscopy of the Au(110) surface near 100 K was performed, and the thermodynamic temperature was determined with an accuracy of about 2 K [11]. In the fitting, the broadening of the spectrum was represented by a linear combination of Gaussian and Lorentzian functions. However, the physical nature of the components in the broadening function has not been assessed adequately. The possibility of asymmetry in the broadening function also needs to be investigated. The broadening function shall be examined in more detail, and it is necessary to derive it directly from the measured photoelectron spectra.

Several scientists in photoelectron spectroscopy have attempted to remove or reduce the effects of the broadening or background by smoothing or deconvolution based on the Fourier transform [12]. There are some reports of deconvolution for the valence bands of gold and copper using XPS, and for the first band of oxygen spectrum using UPS [13,14,15]. In those studies, Gaussian function was assumed as the broadening function and the aims of the research were mainly to derive the intrinsic linewidth. The broadening function was not the main research subject, and its temperature dependence was not sufficiently discussed.

In this study, we tried to derive the broadening functions directly from photoelectron spectra of Au(110) near the Fermi level. The algorithms and conditions in the process of deriving the broadening function and the resulting function contributed to improving the accuracy of thermodynamic temperature measurement based on the photoelectron thermometry. Additionally, the temperature dependence of the broadening function appears to provide information on electron scattering in photoelectron emission.

## 2. Photoelectron Spectra

Photoelectron spectroscopy of a single crystal of Au(110) in the Γ–K direction was performed in a UHV chamber with a Saga Light source, using a high energy resolution electron energy analyzer. Details regarding the measurement method of the photoelectron spectra used in this study are described in our previous article [9]. The sample was cleaned in the UHV chamber by Ar ion sputtering and annealing. The cleanness and crystallinity of the sample were confirmed by X-ray photoelectron spectroscopy (XPS) and low energy electron diffraction (LEED), respectively. Since a spectrum with sufficient intensity was required to improve the fitting accuracy, the photoelectron spectrum was measured with a wider slit width of the analyzer than that in the high energy resolution mode. The measurement was made in 1 meV steps. The sample was mounted on a low-temperature cryostat with a continuous flow of refrigerant liquid, and the temperature was kept constant during measurement. The sample’s temperature was measured by the silicon diode sensor attached to the sample holder during the UPS. The temperatures read when liquid helium and liquid nitrogen were used as the refrigerants were 11 and 99 K, respectively, and the temperature in the absence of refrigerant was 297 K. Band calculation to obtain DOS in Γ–K direction was performed via first-principles calculation using plane wave basis. 

## 3. Computational Methods

The probability of occupation of the electronic states with the energy *ε* in the thermal equilibrium state is described by the FD distribution with a thermodynamic temperature *T*:(1)$$f_{FD}(\varepsilon ,T)=\frac{1}{\exp\left[\left(\varepsilon -E_{F}\right)/k_{B}T\right]+1},$$
where *E*_F_ and *k*_B_ are the Fermi energy and the Boltzmann constant, respectively. The intensity of photoelectron spectrum *I* is regarded as a product of FD distribution and DOS, which is convoluted with a broadening function *f_BR_* with the spectral broadening width Δ*E*.
(2)$$I(E;T,\Delta E)=\int f_{FD}(\varepsilon ,T)\cdot DOS(\varepsilon )\cdot f_{BR}(E-\varepsilon ,\Delta E)d\varepsilon .$$

The spectral broadening width referred to here accounts for not only the limitations of the electron energy analyzer and the light source for spectrum measurement, but also physical properties dependent on the measurement target. Such quantities include sample condition, spread due to the lifetime of an excited state, and electron scattering rate. 

In the spectral region where the DOS is almost constant with respect to the electron energy as in the normal emissions from Au(110), the effect of broadening by DOS is constant. Therefore, the DOS function can be put outside the integral and the normalized photoelectron spectrum *I_np_* can be obtained by dividing the spectrum intensity by DOS: (3)$$I_{np}≃\frac{I(E;T,\Delta E)}{DOS(E)}\propto \int f_{FD}(\varepsilon ,T)\cdot f_{BR}(E-\varepsilon ,\Delta E)d\varepsilon .$$

To derive the broadening function directly from the photoelectron spectrum, the well-known relation between convolution integral and Fourier transform (convolution theorem) can be applied. The Fourier transform of the convolution integral of two functions *f*_1_ and *f*_2_ is equal to the simple product of the Fourier transforms of both functions and is expressed as
(4)$$ℱ\left[f_{1}*f_{2}\right]=ℱ\left[f_{1}\right]\cdot ℱ\left[f_{2}\right].$$

Using these relationships, the function *f*_2_ can be derived from the composite function by dividing both sides of the Equation (4) by the Fourier transform of the function *f*_1_ and inverting the values.
(5)$$f_{2}={ℱ}^{-1}\left[ℱ\left[f_{2}\right]\right]={ℱ}^{-1}\left[\frac{ℱ\left[f_{1}*f_{2}\right]}{ℱ\left[f_{1}\right]}\right].$$

This equation is widely used not only in electron spectroscopy but also in infrared spectroscopy, ultraviolet visible spectroscopy, and X-ray diffraction to obtain an intrinsic linewidth in which the contributions of the instrument’s resolution and the broadening of incident light are reduced. 

The Fourier transform of the normalized photoelectron spectrum is a product of the Fourier transforms of the FD distribution function and the broadening function. The broadening function can be derived as an inverse Fourier transform of the function obtained by dividing the Fourier transform of the normalized spectrum by the Fourier transform of the FD distribution function.
(6)$$f_{BR}\left(E\right)={ℱ}^{-1}\left[ℱ\left[f_{BR}\left(E\right)\right]\right]={ℱ}^{-1}\left[\frac{ℱ\left[I_{ns}\left(E\right)\right]}{ℱ\left[f_{FD}\left(E,T=T_{R}\right)\right]}\right].$$

The thermodynamic temperature *T* of the FD distribution function was assumed to be the reference temperature *T*_R_ measured by a sensor mounted near the sample. To perform the Fourier transform, it is necessary for the numerical data to be continuous periodic data. That is, the first and last data points need to be almost equal. However, the normalized photoelectron spectrum and FD distribution have the initial value of 1 at the lowest energy and the final value of 0 at the highest energy and are discontinuous when regarded as periodic functions. Therefore, the Fourier transform calculation uses the differentials of the photoelectron spectrum and the FD distribution function, which have value 0 at both the beginning and the end.

## 4. Results and Discussion

Before deriving the broadening function from the measured spectrum, theoretical spectra were artificially created, and the derivation of the original broadening function with the proposed method was examined to confirm the effectiveness of this approach and its conditions of application. The theoretical spectrum FDG was generated by convoluting the Fermi–Dirac distribution function with the Gaussian function *G* with the standard deviation *σ* = Δ*E*/2:(7)$$FDG\left(E;T,\Delta E\right)=\int f_{FD}(\varepsilon ,T)\;G(E-\varepsilon ;\Delta E)d\varepsilon .$$

Figure 1 shows the FD distribution at 100 K, the Gaussian function with a width of 30 meV, the composite spectrum FDG, and the differential of the composite spectrum. All data consisted of 1 meV steps, and the centers of the spectra were located at the Fermi energy. 

The resultant functions inversely derived from FDG with various spectral widths at *T* = 100 K are shown in Figure 2. The calculations were performed with three spectrum widths, which were 200, 400, and 600 meV—shown on the left, the center, and the right parts of the figure, respectively. The red lines in the upper, middle, and lower figures are the original Gaussian functions convolved as the broadening functions with the widths, Δ*E*, of 10, 30, and 50 meV, respectively. 

In the case of Figure 2a–c, on the left side with a spectrum width of 200 meV, the peak intensities of the derived functions are much higher than the original Gaussian function and diverged at the origin. Instead, the functions derived from the FDG functions with a spectrum width of 400 meV shown in Figure 2d–f, reproduced the original Gaussian functions with broadening widths of 10 and 30 meV. However, in the spectrum with a broadening width of 50 meV, the derived broadening function was still oscillating and noisy. In Figure 2g–i with a spectrum width of 600 meV, it is clear that the results reproduced the original Gaussian function for every broadening width.

From the results, it was found that an appropriately wide spectral width is required to derive a broadening function using this method that is in accordance with the broadening width, Δ*E*. In the same analysis, for a spectrum with a broadening width of 50 meV or less at the temperature of 100 K, the broadening function can be derived with a spectral width of 600 meV. Similarly, the temperature dependence of the required spectral width for deriving the broadening function was investigated. The photoelectron spectrum requires the spectral width of 400 meV to derive a broadening function with a broadening width of 50 meV or less at a temperature of 10 K, whereas the photoelectron spectrum at a temperature of 300 K requires the spectral width of 1200 meV. This can be understood from the behavior of the FD distribution with respect to temperature. 

Similar tests were performed for the asymmetric broadening function. A broadening function was derived by the above method from a composite function obtained by combining right and left half-Gaussian functions with different half-widths of half maximum and convoluting with the FD distribution function. It was confirmed that the broadening function can be reproduced even in the asymmetric function for the above conditions.

As the next step, the measured photoelectron spectra of Au(110) normalized by DOS and the FD distribution functions at three temperatures are shown in Figure 3a,c,e. The differentials of the normalized photoelectron spectra and the FD distribution function are shown in Figure 3b,d,f. The differentiator was the difference between adjacent data. According to the calculations discussed above, spectrum widths of 400, 600, and 1200 meV are required for spectra at 11, 99, and 297 K, respectively, to derive the broadening function from the FDG function with a broadening width of 50 meV or less. Additionally, the convolution relation (Equation (4)) needs continuous periodic data. However, the intensity of the FD distribution function and the photoelectron spectrum are step-like functions from 1 to 0. Therefore, the photoelectron spectra were differentiated, and both ends were complemented with zeros to obtain continuous periodic data.

An attempt to derive the broadening function was made by dividing the Fourier transform of the differentiated photoelectron spectrum by the Fourier transform of the differentiated FD distribution function and performing an inverse Fourier transform, as mentioned in Section 3. However, the calculated results were scattered and did not show a distribution that seemed to be a broadening function. This is because the Fourier transform of the FD distribution function approaches 0 at a high-order component, but the Fourier transform of the photoelectron spectrum does not converge to 0 due to its noise. Therefore, we tried to reduce noise by replacing high-order components of the amplitude spectrum after the Fourier transform with zero, as is done in band-pass filtering. 

The substitution of zero was applied based on the amplitude spectrum after Fourier transforming the photoelectron spectrum. The amplitude spectra before and after zero substitution are shown in Figure 4. All the high-order components were replaced with the value zero above the point where the amplitude spectrum was not monotonically decreasing. For example, in the spectrum at 99 K (in Figure 4b), the amplitudes of the 18th and higher orders were regarded as noise components in the measurement and were replaced with the value zero. This is a so-called low-pass filter, which is equivalent to fitting the differentiated photoelectron spectrum with a 17th order Fourier series. The broadening function then was derived from the noise-reduced data.

The broadening functions derived from photoelectron spectra of Au(110) measured at three temperatures, 11, 99, and 297 K are shown in Figure 5a–c, respectively. We succeeded in deriving the shape of the distribution function, which seems to be the broadening function at the center. The derived broadening functions were symmetric and had a form of Gaussian distribution with the standard deviation *σ* = Δ*E*/2. Δ*E* = 14.1, 17.5, and 37.8 meV at 11, 99, and 297 K, respectively. 

It was found that the width of the broadening function increased as the temperature increased. It is considered that thermal scattering of electrons in the photoemission process increases the broadening width of the spectrum. The theoretical study of electron scattering in photoemission has been performed for a long time [16]. Temperature dependence of photoemission from quantum-well states has been examined [17]. The peak width changes were explained in terms of electron–phonon coupling. It is likely that the lifetime of the excited state depends on the thermodynamic temperature. The temperature-dependent linewidth of the surface state of Au(100) was also measured [18]. It was argued that the temperature dependence was due to the presence of thermally excited defects on the surface. In the present study, the broadening functions were determined in three temperature regions. It can be anticipated that these phenomena will be elucidated in more detail by determining the broadening function of photoemission at sufficient measurement points in a wide temperature range. 

To estimate how much the uncertainty of the value of temperature influences the derived broadening function, similar calculations were performed with different reference temperatures. It was found that when the reference temperature was changed by 1 K, the width of the derived Gaussian function of the broadening function changed by about 0.2 meV at all three temperatures of 11, 99, and 297 K.

In the previous study, a linear combination of Gaussian and Lorentzian functions was used as the broadening function when determining the thermodynamic temperature [9]. There, DOS was regarded as constant, and the photoelectron spectrum was not normalized using it. In the present study, the shape of the Gaussian function was derived as the broadening function from the photoelectron spectrum normalized with the DOS, and the component of the Lorentzian function was not derived. Therefore, the FD distribution with only the Gaussian function, as the broadening factor was fitted to the normalized photoelectron spectrum. As a result, we confirmed that the fitting can be performed successfully with only the Gaussian as the broadening function. It was also concluded that the normalization of the photoelectron spectrum by DOS is essential even for the determination of thermodynamic temperature from the photoelectron spectrum. Furthermore, when fitting was performed in a noise-reduced spectrum at 99 K using the Fourier transform, it was confirmed that the standard deviation of the determined temperature was suppressed to less than 1 K.

## 5. Conclusions

The broadening function was derived from the photoelectron spectrum of the Au(110) surface near the Fermi level using the relationship between the Fourier transform and the convolution integral. It was found that an adequate spectral width is necessary to derive the broadening function using the Fourier transform. Moreover, the noise could be reduced by using the Fourier transform when deriving the broadening function. The derived broadening function is in the form of a Gaussian function, which depends on the thermodynamic temperature of the sample, and becomes wider at higher temperatures.

In the present research, we analyzed the broadening of the spectrum that is indispensable for determining thermodynamic temperatures. Normalization of the photoelectron spectrum via the calculated DOS enhanced the reliability in fitting. In addition, the noise reduction using the Fourier transform in this study increased the precision of the determination. The present results improve the accuracy of determining the thermodynamic temperature from the photoelectron spectrum.

## Figures and Tables

**Figure 1 sensors-21-05969-f001:**
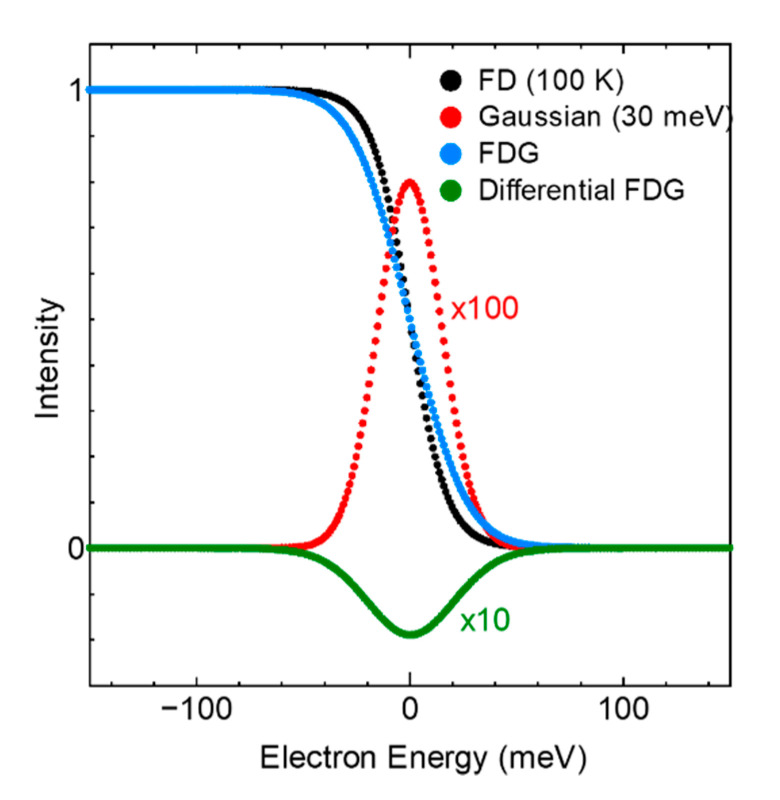
FD distribution of 100 K (black), Gaussian function with a width of 30 meV (red); enlarged100 times and 10 times, respectively.

**Figure 2 sensors-21-05969-f002:**
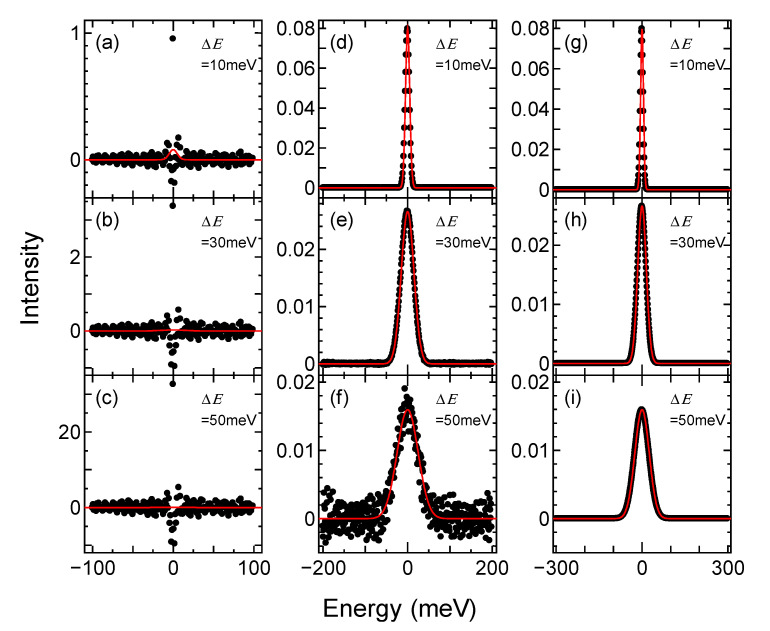
The broadening functions derived from FDG function of *T* = 100 K (black dots) and the convoluted original Gaussian functions (red lines). The spectrum widths *W*sp and the broadening widths of the original Gaussian functions Δ*E* (**a**) *W*sp = 200 meV, Δ*E* = 10 meV, (**b**) *W*sp = 200 meV, Δ*E* = 30 meV, (**c**) *W*sp = 200 meV, Δ*E* = 50 meV, (**d**) *W*sp = 400 meV, Δ*E* = 10 meV, (**e**) *W*sp = 400 meV, Δ*E* = 30 meV, (**f**) *W*sp = 400 meV, Δ*E* = 50 meV, (**g**) *W*sp = 600 meV, Δ*E* = 10 meV, (**h**) *W*sp = 600 meV, Δ*E* = 30 meV, (**i**) *W*sp = 600 meV, Δ*E* = 50 meV.

**Figure 3 sensors-21-05969-f003:**
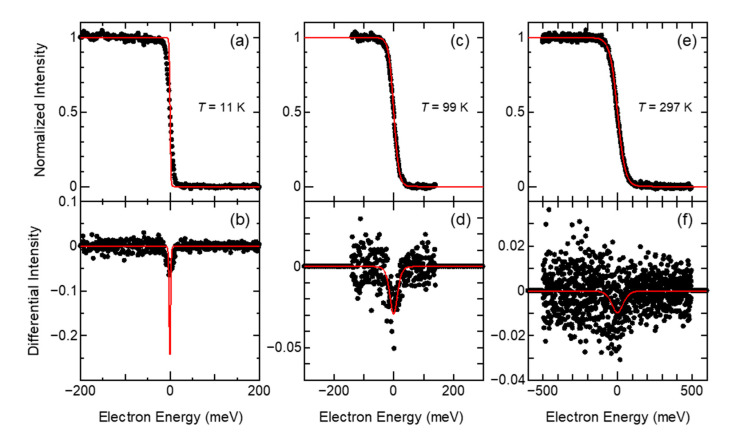
Normalized photoelectron spectra of Au(110) (upper row) and the differential photoelectron spectra (lower row) measured at 11 K (**a**,**b**), at 99 K (**c**,**d**), and at 297 K (**e**,**f**). The FD distribution function at each temperature is shown overlaid on each spectrum (upper row, red), and the differential FD distribution is shown overlaid on each differential spectrum (lower row, red).

**Figure 4 sensors-21-05969-f004:**
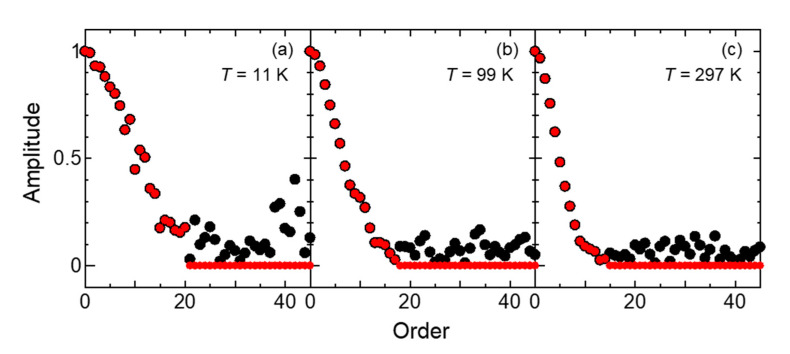
Fourier-transformed amplitude spectra before zero substitution (black) and after zero substitution (red) for the spectra measured at 11 K (**a**), 99 K (**b**), and 297 K (**c**).

**Figure 5 sensors-21-05969-f005:**
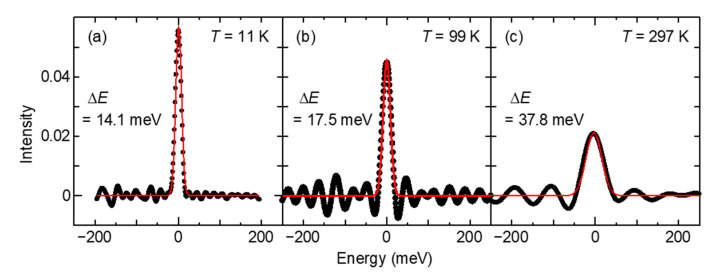
Broadening functions derived from photoelectron spectra of Au(110) measured at liquid helium cooling temperature, *T* = 11 K (**a**), at liquid nitrogen cooling temperature, *T* = 99 K (**b**), and at room temperature, *T* = 297 K (**c**). Each red line shows a fitting result using the Gaussian function with the broadening width Δ*E* shown.

## Data Availability

Data collected through research in the paper are available on request from the corresponding authors.

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
