# Peer review of "Temperature-Dependent Broadening of the Ultraviolet Photoelectron Spectrum of Au(110)"

_sensors, 2021, doi:10.3390/s21175969_

Round 1
Reviewer 1 Report
I don't think the theme of this article is consistent with the journal. I advise reject this paper. I have some comment as follow: 1. The introduction is not good, which is not logic. 2. There are a few pictures (only three figures). 3. There are no pictures for gold. 4. The references are old and little. 5. The number of the temperature is little (Figure 3)Author Response
Dear Reviewer,
The authors appreciate the comments from the reviewer.
The authors followed the reviewer's comments and revised the manuscript for sensor journal readers. Below are the responses to the reviewer's comments.
1. The introduction has been revised.
2. The authors added two figures to the manuscript.
3. The authors can't find any meaning in posting a picture of the sample.
4. Currently, there are very few references related to the present research.
5. There is a limit to the types of refrigerants that can be used for cooling in measurement, and it is difficult to measure at many temperature points. However, the authors consider that three temperature points are sufficient to see the temperature-dependent tendency.
Sincerely.

Reviewer 2 Report
In the paper authors propose a new method of evaluation photoelectron spectrum energy broadening based on Fourier transform formalism. Present paper is a continuation, improved version of previous studies performed by the authors aiming at photoelectron thermometry.
Basically, the subject of the article (photoelectron thermometry) is studied solely by the authors. I have not found any recent papers (dated within last decade) devoted to this subject. There is no feed back to their last paper (which is dated 2017) on the subject. I have checked on the Web of Knowledge - the paper has not been cited at all. Therefore in my opinion the interest in the community would be rather poor. On the other hand the paper is written clearly. I had only one remark, that there should be more results presented in order to make the conclusions more reliable.
Paper written clearly but not very sound.
As for the presented results, the data given in Fig. 2a are the same as those shown in the paper [9]: Jpn. J. Appl. Phys. 2017, 56, 048004. The noise in Fig. 2b is pretty high. It would be fine if the authors presented similar data for 11K and 297K as well.
Author Response
Dear Reviewer,
The authors appreciate the comments from the reviewer.
The authors followed the reviewer's comments and revised the manuscript for sensor journal readers. The authors added two more figures and presented results for 11 K and 297 K.
Sincerely.

Reviewer 3 Report
The paper describes an interesting method to measure surface temperature by UV photoelectron spectroscopy. The idea and method is interesting for the readers of the sensor journal. However, the paper is quite comprehensive and needs some überarbeitung, to explain the method more clearly to the readers to the sensor journal, which might not be familiar with this method. Additionally it would be good to rank the present results with results already obtained with this method and what is the new and/or innovative aspect of this paper. Some other remarks and questions are given in the attached file

Author Response
Dear Reviewer,
The authors are very grateful for the reviewer's comments and suggestions for specific corrections.
The authors followed the reviewer's suggestions and revised the manuscript to make it as understandable as possible to the readers of the sensor journal.
This paper is not a research that determines the thermodynamic temperature itself, but one that analyzes broadening, which is important for determining the thermodynamic temperature, so it is difficult to compare it with results already obtained.
Sincerely.

Round 2
Reviewer 1 Report
This article lacks experimental support. The research orientation is more inclined to electronic. I suggest this paper to submit other journals. The reference is out-dateAuthor Response
Please see the attachment.

Reviewer 2 Report
I appreciate that the article has been improved following my remarks.
In the revised version there is a promise (in Conclusions):
It is considered that the present results improve the accuracy of determining the thermodynamic temperature from the photoelectron spectrum.
With reference to this statement I have a question what is the accuracy of the temperature measurement with the use of this very model of broadening? In their earlier papers, the authors have shown what was the difference between the temperature determined with their model and real temperature [7], [10], [11]. This let them conclude about the correctness of the model and assumptions they made. In this article such piece of information is missing.
Reviewer 3 Report
Thanks, all issues have been addressed and the paper is fine now.
